



# Risk of crop failure due to compound dry and hot extremes estimated with nested copulas

Andreia Filipa Silva Ribeiro[1,2], Ana Russo[2], Célia Marina Gouveia[2,3], Patrícia Páscoa[2,3,4], and Jakob Zscheischler[1,5]

[1]Climate and Environmental Physics, University of Bern, Sidlerstrasse 5, 3012 Bern, Switzerland
[2]Instituto Dom Luiz (IDL), Faculdade de Ciências, Universidade de Lisboa, Campo Grande, 1749–016, Lisboa, Portugal
[3]Instituto Português do Mar e da Atmosfera, Lisboa, Portugal
[4]Environmental Physics Laboratory (EPhysLab), University of Vigo, Ourense, Spain
[5]Oeschger Centre for Climate Change Research, University of Bern, Bern, Switzerland

**Correspondence:** Andreia F. S. Ribeiro (afsribeiro@fc.ul.pt)

**Abstract.** Drought and heat events stress agricultural systems and may threaten food security. The interaction between co-occurring drought and hot conditions is often particularly damaging to crop's health and may cause crop failure. In this context, traditional univariate analyses may not be adequate for reliable risk assessment of crop failure associated with compound hazards. Climate change exacerbates such risks due to an increase in the intensity and frequency of dry and hot events in many land regions. Here we model the trivariate dependence between spring maximum temperature, spring precipitation and wheat and barley yields, respectively, over two province clusters in Spain with nested copulas. Based on the full trivariate joint distribution, we (i) estimate the impact of compound hot and dry conditions on wheat and barley loss and (ii) estimate the additional impact due to compound hazards compared to individual hazards. We find that crop loss increases when drought- or heat-stress aggravates to compound dry and hot conditions and that an increase in the severity of compound conditions leads to larger damages. For instance, compared to moderate drought only, compound dry and hot conditions increase the likelihood of crop loss by 8 to 11% while when starting with moderate heat, the increase is between 19 to 29% (depending on the cereal and region). This findings suggest that the likelihood of crop loss is driven primarily by drought stress than by heat stress, suggesting that drought plays the dominant role in the compound event, that is, drought stress does not require to be so extreme as heat stress to cause a similar damage. Furthermore, when compound dry and hot conditions aggravate from moderate to severe or extreme stress, crop loss probabilities increase 5 to 6% and 6 to 8% respectively (depending on the cereal and region). Our results highlight the additional value of a trivariate approach for the estimating the compounding effects of dry and hot extremes on of crop failure risk. Therefore, this approach can effectively contribute to design management options and guide the decision-making process in agricultural practices.

## 1 Introduction

The assessment of the adverse social, economic and environmental impacts associated with a combination of multiple climate hazards have recently become a focus of high interest (Leonard et al., 2014; Zscheischler et al., in press). Such compound events





often lead to larger impacts compared to when hazards occur separately (Zscheischler et al., 2018). For instance, compound dry and hot conditions reduce carbon uptake more strongly compared to the sum of the individual hazards (Zscheischler et al., 2014). Dry and hot conditions often co-occur. For instance in Europe, the extreme 2003, 2010 and 2018 heatwaves were

accompanied by strong soil moisture deficits (Bastos et al., 2014; Schumacher et al., 2019; Buras et al., 2020). In 2010, the compound event was particularly strong in Russia (Schumacher et al., 2019), while in 2003 the extreme drought and heatwave affected mostly central Europe, extending to west Mediterranean countries like Portugal and Spain (Garcia-Herrera et al., 2010), with critical consequences in several sectors. In 2010, widespread crop yield declines and failures occurred over the major grain producing regions of Russia, northeastern Ukraine, and northwestern Kazakhstan (Loboda et al., 2017). Previously,

the shortages in crop yields in 2003 have also caused major financial losses in the agricultural sector, and when compared to the previous year, the cereal productions in European Union (EU) have decreased 23 million tonnes (COPA-COGECA, 2003). The decline in the harvests was both in quantity and quality, as was the case of winter cereals whose maturation was accelerated due to compound extreme dry and hot conditions, forming grains with insufficient water content (COPA-COGECA, 2003). The 2018 event strongly impacted pastures and arable land north of the Alps (Buras et al., 2020). As the occurrence of climate

extremes such as heatwaves, droughts and compound dry and hot events is expected to increase in intensity and frequency in many land regions due to climate change (IPCC, 2012; Zscheischler and Seneviratne, 2017), associated adverse impacts such as widespread harvest failures threatening global cereals supplies may also increase.

Among the panoply of multivariate approaches applied to assess the impacts of multiple climate hazards, the use of copulas has become quite popular in studies focused on analysing the social, environmental and economic risks associated with adverse

climate conditions (Bokusheva et al., 2016; Gaupp et al., 2019; Madadgar et al., 2017; Ribeiro et al., 2019b, a; Zscheischler et al., 2017). With copulas nonlinear dependency structure can be modelled, which offers more flexibility and possibly a more adequate fit for different dependence types in the extremes. (Durante and Sempi, 2015; Nelsen, 2006; Salvadori and De Michele, 2007; Salvadori et al., 2016). Among all types of copulas described in the literature, the popularity of the class of elliptical copulas comes from the fact that they derive from well known distributions associated to the widely used Pearson's

correlation, but the elliptical dependence is only able to capture radial symmetry and the respective mathematical expressions do not have a closed form. One of the copula classes that overcomes this drawback is the Archimedean, which have a simpler mathematical form and can capture different kinds of tail dependence and radial symmetry or asymmetry.

Archimedean copulas (AC) are exchangeable, which means that the copula is the same if we permute the respective margins. For the bivariate case this may not be a limitation, but as the number of dimensions increase, it is unlikely that exchanging

across the involved variables allows for the 'true' dependence structure to be well-defined. To avoid exchangeability, nested Archimedean copulas (NAC) have been proposed (Okhrin and Ristig, 2014), also referred to as hierarchical Archimedean copulas (HAC), obtained by nesting lower dimensional Archimedean copulas into each other and/or with marginal distributions. Okhrin and Ristig (2014) introduced NACs where all copulas belong to the same family with a nesting condition that requires decreasing dependence strength from the highest to the lowest hierarchical level. Here we make use of this NAC approach,

taking advantage of the balance between flexibility (modelling different types of dependence structures) and usability in higher dimensions (limiting the number of parameters).





The present work aims to identify how risks associated with compound dry and hot conditions affect wheat and barley yields over two clusters of provinces in Spain based on the trivariate dependence between precipitation, maximum temperature and yields using a NAC approach. In particular, we are interested in quantifying the additional risk associated with compound dry

and hot conditions compared to only dry or only hot conditions. Wheat and barley are chosen as they are two of the major rainfed crops in the Iberian Peninsula (Peña-Gallardo et al., 2019; Vicente-Serrano et al., 2006). Moreover, we here build on prior work which has estimated wheat and barley losses in the same area, but related to a single hazard, namely droughts (Ribeiro et al., 2019a, b).

Using NACs, we estimate the conditional probabilities of crop loss under different severity levels of dry and hot conditions

based on the full trivariate joint distribution. We focus on annual wheat and barley yield data at the sub-national scale, thus overcoming drawbacks related to assessing climate related crop risks at the national scale. Based on the proposed approach we (i) characterize the dependence structures between the dry and hot conditions and the crop yields; (ii) estimate the conditional probability of crop loss under different compound dry and hot severity levels; and (iii) evaluate how much the compound dry and hot conditions increase the risk of crop failure in comparison to the individual hazards.

## 2  Data and methods

### 2.1  Crop yield data

Wheat and barley yields were obtained for 9 provinces in Spain from the Spanish Agriculture, Fishing and Environment Ministry (available at https://www.mapa.gob.es/es/estadistica/temas/publicaciones/anuario-de-estadistica/, last access on 9 November 2019). The data were assembled in two clusters of provinces (Figure 1) which are dominated by rainfed agricultural prac-

tices considering the non-irrigated arable land classification from CORINE Land Cover dataset based on an earlier clustering (Ribeiro et al., 2019c, b). Figure 1 shows the Iberian provinces with $< 50\%$ agricultural pixels colored in white, the provinces with $> 50\%$ agricultural pixels coloured with the respective agricultural CORINE classes and the selected two clusters of contiguous provinces dominated by rainfed agriculture delineated in bold black contours.

Crop yields were obtained as the ratio between production and harvested area during the period of 1986–2016. We computed

crop yield anomalies by removing longer term trends based on locally estimated scatterplot smoothing (LOESS, a method for local regression) to account for yield increases due to technological development (Ben-Ari et al., 2016). We pooled crop yields from the provinces over each cluster, resulting in samples sizes $N_1 = 155$ for Cluster 1 (30 years of annual data over five provinces) and $N_2 = 124$ for Cluster 2 (30 years of annual data over four provinces). Pooling time series greatly expands the sample size allowing greater robustness in three-dimensional statistical analysis that otherwise would be compromised. This

type of assessment is a compromise between the use of a sub-national resolution of crop data and the sample size to evaluate the number of cases of simultaneous occurrence of dry and hot conditions.



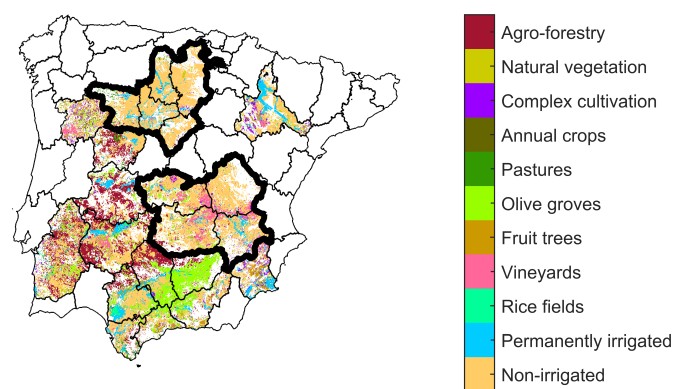

**Figure 1.** Iberian provinces dominated by agricultural land use ($> 50\%$ agricultural pixels belonging to all agricultural CORINE classes, see legend) according to the CORINE Land Cover dataset and respective categories. The contiguous provinces dominated by rainfed practices ($> 50\%$ non-irrigated pixels in yellow) are delineated in bold black contours and grouped in two clusters. Northern region (Cluster 1) provinces: Burgos, Palencia, Segovia, Valladolid, and Zamora. Southern region (Cluster 2) provinces: Albacete, Ciudad Real, Cuenca, and Toledo.

## 2.2 Weather data

The vegetative cycle of the winter crops in Spain is mainly driven by precipitation and temperature: sowing occurs around autumn, followed by the vegetative phase in winter, reproductive phase in spring (when vegetation is photo-synthetically more active) and crop harvest occurs in the early summer. Therefore, monthly precipitation (P) and monthly maximum temperature (Tmax) were extracted from the Climate Research Unit (CRU) TS4.01 dataset (Harris et al., 2014) spanning the same time period. We used 3-monthly means of Tmax and 3-monthly means of P during spring ($P_{MAM}$ and $Tmax_{MAM}$, respectively), which was identified as the most sensitive time period for crop yield based on correlation analysis (Figure 2). This selection of climate variables allows to maximize the dependence between climate conditions and yields as also shown by previous work based on the same data (Ribeiro et al., 2019c).

We considered three severity levels of dry and/or hot conditions: Moderate (+), Severe (++) and Extreme (+++) based on percentile thresholds as shown in Table 1. Besides these three severity levels, we further considered all combinations of 10 categories of severity levels of dry and hot conditions exceeding the $50^{th}$ to $5^{th}$ and $50^{th}$ to $95^{th}$ percentiles for $P_{MAM}$ and $Tmax_{MAM}$, respectively. We further considered the $20^{th}$ percentile of the crop anomaly time-series as lower exceedance threshold for crop failure (Ben-Ari et al., 2016; Ribeiro et al., 2019a, b).





**Table 1.** Categories of severity levels of dry and hot conditions based on $P_{MAM}$ and $Tmax_{MAM}$ percentiles.

|  | Moderate (+) | Severe (++) | Extreme (+++) |
|---|---|---|---|
| dry | $P_{MAM} \leq 20^{th}$ percentile | $P_{MAM} \leq 10^{th}$ percentile | $P_{MAM} \leq 5^{th}$ percentile |
| hot | $Tmax_{MAM} \geq 80^{th}$ percentile | $Tmax_{MAM} \geq 90^{th}$ percentile | $Tmax_{MAM} \geq 95^{th}$ percentile |

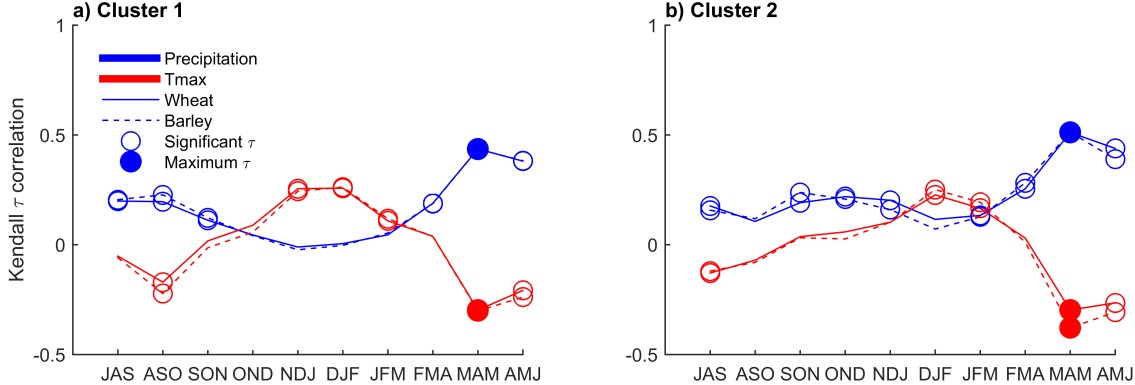

**Figure 2.** Kendall correlation $\tau$ between three-monthly means of maximum temperature (Tmax, red) and precipitation (blue) with wheat (filled lines) and barley (dashed lines) yield anomalies, respectively. Correlations were computed during the crop growing period (September to June) over 1986-2016 for Cluster 1 (a) and 2 (b) (Figure 1). The months on the x-axis denote the end-month of the averaging period. Circles indicate statistically significant correlations at $\alpha = 0.05$. The strongest correlation (positive or negative) is denoted by filled circles ($P_{MAM}$ and $Tmax_{MAM}$).

## 2.3 Modelling trivariate distributions with nested Archimedean copulas

We model the trivariate relationship between temperature, precipitation and crop yields with nested copulas. Consider a vector of crop yield annual anomalies $Y$ and the climate variables $X_1 = P_{MAM}$ and $X_2 = Tmax_{MAM}$ with marginal cumulative distribution functions (CDF) $F_Y$, $F_{X_1}$ and $F_{X_2}$, respectively. We aim to estimate and compare three conditional cumulative distribution functions (CDFs) with the scalars $x_1^*$ and $x_2^*$ corresponding to the dry and hot thresholds, respectively:

$$F_{Y|X_1}(Y|X_1 = x_1^*) = P(Y \leq y | X_1 \leq x_1^*) \tag{1}$$

$$F_{Y|X_2}(Y|X_2 = x_2^*) = P(Y \leq y | X_2 \geq x_2^*) \tag{2}$$

$$F_{Y|X_1,X_2}(Y|X_1 = x_1^*, X_2 = x_2^*) = P(Y \leq y | X_1 \leq x_1^*, X_2 \geq x_2^*) \tag{3}$$

With the above equations we can estimate the agricultural impacts under dry conditions $F_{Y|X_1}$ (Equation 1), under hot conditions $F_{Y|X_2}$ (Equation 2) and under compound dry and hot condition $F_{Y|X_1,X_2}$ (Equation 3), respectively. In other words, if the compound dry and hot conditions cause more damage than the individual hazards, it is expected that $F_{Y|X_1,X_2}$





suggests higher probabilities of crop loss (i.e., $y = y^*$ for a low $y^*$) than $F_{Y|X_1}$ or $F_{Y|X_2}$. Furthermore, we can study the relative role of $P_{MAM}$ and $Tmax_{MAM}$ in crop loss with Equations 1 and 2.

To compare the additional impact of compound dry and hot conditions with the impacts caused by the individual hazards,

Equations 1, 2 and 3 are used to estimate

$$\text{Relative change from drought-stress} = \frac{F_{Y|X_1=x_1^*,X_2=x_2^*}(0.2) - F_{Y|X_1=x_1^*}(0.2)}{F_{Y|X_1=x_1^*}(0.2)} \tag{4}$$

$$\text{Relative change from heat-stress} = \frac{F_{Y|X_1=x_1^*,X_2=x_2^*}(0.2) - F_{Y|X_2=x_2^*}(0.2)}{F_{Y|X_2=x_2^*}(0.2)}, \tag{5}$$

where 0.2 is the threshold of crop loss ($y^*$) corresponding to the $20^{\text{th}}$ percentile of the crop yields. These changes can be estimated for different severity levels of dry ($x_1^*$) and hot ($x_2^*$) conditions.

Following the theorem of Sklar (1959) we can decompose a multivariate probability distribution into its marginals and a copula $C$ which describes the dependence structure between the margins. To estimate the multivariate distribution $P(Y, X_2, X_3)$, the respective copula $C$ is fitted, which is then a joint CDF whose marginal distributions are uniform in the interval $[0,1]$ (Durante and Sempi, 2015; Nelsen, 2006; Salvadori and De Michele, 2007). Transforming the margins to uniform variables through their CDFs, that is, $u_1 = F_Y$, $u_2 = F_{X_1}$ and $u_3 = F_{X_2}$, the trivariate CDF can be written as (Sklar, 1959):

$$F(u_1, u_2, u_3) = C(u_1, u_2, u_3). \tag{6}$$

Within the copula families, AC are extensively used due to their flexibility and applicability to a variety of tail dependence structures, as well as their analytical tractability. AC can be written in terms of the respective generator function $\varphi$, e.g. for the three-dimensional case:

$$C(u_1, u_2, u_3; \theta) = \varphi_\theta(\varphi_\theta^{-1}(u_1) + \varphi_\theta^{-1}(u_2) + \varphi_\theta^{-1}(u_3)) \tag{7}$$

Due to the symmetry of bivariate AC, the above trivariate form can be expressed in terms of NAC or HAC, where two of the margins are coupled by their bivariate copula,

$$C(u_1, u_2, u_3; \theta_{12}; \theta_1) = C_1(C_{12}(u_1, u_2; \theta_{12}), u_3; \theta_1) \tag{8}$$

Equation 8 can also be expressed in terms of the other possible pair copulas $C_{13}(u_1, u_3; \theta_{13})$ and $C_{23}(u_2, u_3; \theta_{23})$ that are coupled with $u_2$ and $u_1$ by $C_2$ and $C_3$, with expressions $C_2(C_{13}(u_1, u_3; \theta_{13}), u_2; \theta_2)$ and $C_3(C_{23}(u_2, u_3; \theta_{23}), u_1; \theta_3)$, respectively.

Most structures of NAC require decreasing parameters from the inner to the outer hierarchical level to attain a properly fitted copula. As for most ACs, the larger the parameter the stronger the dependence, this means that most structures of NAC require that the marginal copulas in the inner level should correspond to the pair with the strongest dependence, i.e., satisfying $\theta_{12} \geq \theta_1$ in the case of Equation 8. This requirement applies to NAC with generators from the same family, providing a flexible estimation of the NAC, which allows for specifying the full distribution with at most $d-1$ parameters, where $d$ is the number

of copula dimensions or marginal distributions (Okhrin and Ristig, 2014).





In our study we focus on a total of four Archimedean families that capture different kinds of joint dependence structures: Clayton, Gumbel, Frank and Joe. The Clayton, Gumbel and Joe copulas describe an asymmetrical tail behaviour, while the Frank copula, in a similar way to the Gaussian copula, captures joint symmetric dependence. While Gumbel and Joe copulas can represent upper tail dependence, Clayton copulas can represent lower tail dependence. The estimation of the copula parameters

is based on maximum likelihood based on the R package HAC (Okhrin and Ristig, 2014) .

The main steps of the trivariate approach used in this study can be summarized as follows (Okhrin and Ristig, 2014). First, the marginal distributions $u_1$, $u_2$ and $u_3$ are estimated non-parametrically by simple ranking, a common approach for copula modelling. Afterwards, the fit of bivariate copula models is performed to every pair of variables to estimate $C_{\theta_{12}}$, $C_{\theta_{13}}$ and $C_{\theta_{23}}$. For each pair, the copula selection is performed based on the Akaike's information criterion (AIC) and checking the

goodness-of-fit by comparing the empirical copula based on the Cramer-von Mises distance ($Sn$). The bivariate copula with the strongest dependence, with the lowest AIC and the lowest $Sn$, is selected to define the structure of the NAC. Afterwards, the marginal distribution that is not part of the selected bivariate copula is joined and the parameter of the upper level copula of the same family is estimated (Equation 8). As a final step, the estimated NAC with two parameters is compared with the same Archimedean family with one parameter (Equation 7) in terms of the AIC, which penalizes the number of estimated

parameters.

### 2.4 Diagnostics and uncertainties in the estimation procedure

The visual diagnostics of the quality of the selected models are performed analogously to a QQ-plot by comparing the empirical estimate of the Kendall function (cumulative distribution of the copula) with the theoretical estimate of the Kendall function based on the selected parametric trivariate copulas (Okhrin and Ristig, 2014).

Best estimates of all conditional probabilities (i.e., Equations 1-5) are estimated by drawing $N = 100,000$ samples from the fitted trivariate copula. Due to the negative dependence between $\text{Tmax}_{\text{MAM}}$ and both crop yields and $\text{P}_{\text{MAM}}$, we inverted the margins of $\text{Tmax}_{\text{MAM}}$ for copula modelling.

Uncertainties of the statistical modelling are estimated by repeated sampling (10,000 times) of the fitted model with sample sizes equal to the number of observations (i.e., $N_1$ in the case of Cluster 1 and $N_2$ in the case of Cluster 2). From these

samples, 95% confidence intervals of Kendall's rank correlation are estimated and compared with the observed pairs $(u_1, u_2)$, $(u_1, u_3)$ and $(u_2, u_3)$. This validation step intends to verify if the generated pairs of copula-based samples preserve the level of dependence found in the observations. Furthermore, this approach is used to estimate uncertainties related to the conditional probabilities (Equations 1-5).

### 3 Results

In both cereals and both clusters the most dependent pair of variables corresponds to crop yields and $\text{P}_{\text{MAM}}$, hence the pair of variables $u_1, u_2$ defines the optimal NAC structure (Figure 3). Results for all possible variable pairs and the respective bivariate copulas are shown in Table A.1.





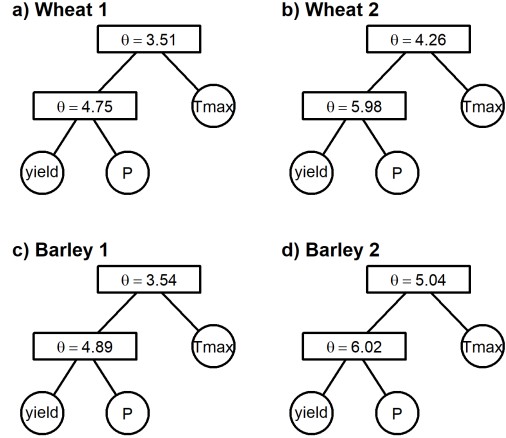

**Figure 3.** Structure and respective parameters of the selected nested Frank models $C_1(C_{12}(u_1, u_2; \theta_{12}), u_3; \theta_1)$ to model the trivariate joint distributions between crop yields, $P_{MAM}$ and $Tmax_{MAM}$. (a) Wheat in Cluster 1. (b) Wheat in Cluster 2. (c) Barley in Cluster 1. (d) Barley in Cluster 2.

Once the bivariate copula $C_{12}(u_1, u_2)$ of yields and $P_{MAM}$ is known, the NAC models are constructed (Table 2). The Frank copula provides the best fit of $C_{12}(u_1, u_2)$ (Table A.1) for both cereals and both clusters and thus the parameters of the trivariate nested copulas are all from the Frank family. Nevertheless, despite Frank being the best family to characterize the nested copulas, we also constructed NAC models with Gumbel, Clayton and Joe copulas for comparison, as well as trivariate Archimedean copulas with one parameter where we selected the best structure between one-parameter and two-parameter AC copulas via the AIC (Table 2). In all but one case the NAC models with Frank copulas is the best model. The only exception is barley in Cluster 2 whose AIC of $C_\theta(u_1, u_2, u_3)$ is slightly lower than the AIC of $C_{\theta_1}(u_3, C_{\theta_{12}}(u_1, u_2))$ (Table 2). Nevertheless, in terms of Cramer-von Mises distance ($Sn$) the nested copula is the closer to the empirical trivariate copula. For this reason, we modelled the trivariate joint distribution based on nested Frank copulas for all cases. For all fitted models, the empirical cumulative distribution corresponds well to the theoretical cumulative distributions (Figure 4). Bivariate dependencies as measured by Kendall's $\tau$ are captured well by the fitted models (Figure 5 for wheat, Figure A.1 for barley).

The cumulative conditional probabilities of yield under moderate (+), severe (++) and extreme (+++) compound dry and hot conditions demonstrate that the probability of crop loss increases with the severity of compound dry and hot conditions for both clusters and both cereals (Figure 6a-d). Moreover, the likelihood of crop loss is higher in Cluster 2 for both cereals, particularly in the case of barley. Under extreme dry and hot conditions (+++dry+++hot, purple), the likelihood of crop loss is 68% and 71% for wheat and barley, respectively, in Cluster 2, in contrast to 62% and 63% in Cluster 1 (Figure 6e, purple bars). In addition, the differences in crop loss are higher between moderate (+dry+hot) and severe (++dry++hot) conditions compared to the differences between severe and extreme (+++dry+++hot) conditions. More precisely, when the compound dry and hot conditions aggravate from moderate to severe stress, crop loss increases 5 to 6% and when the compound dry and hot





**Table 2.** Trivariate Archimedean copulas (AC) parameters ($\theta$) with nested structure with two-parameters $C_1(C_{12}(u_1, u_2; \theta_{12}), u_3; \theta_1)$ and with one-parameter $C(u_1, u_2, u_3; \theta)$ and respective Akaike's Information Criteria (AIC) and Cramer-von Mises distance (Sn). Fit based on maximum pseudo-likelihood (Gumbel (G), Clayton (C), Frank (F) and Joe (J) copulas). Smaller values of AIC and Sn indicate the selected copula for each cereal and cluster (bold).

| | | | Cluster 1 | | | | | Cluster 2 | | | |
|---|---|---|---|---|---|---|---|---|---|---|---|
| | | | G | C | F | J | | G | C | F | J |
| Wheat | $C(u_1, u_2, u_3; \theta)$ | $\theta$ | 1.41 | 0.66 | 3.22 | 1.53 | $\theta$ | 1.53 | 0.75 | 3.88 | 1.72 |
| | | AIC | -74.16 | -79.89 | -99.02 | -49.16 | AIC | -89.12 | -74.67 | -106.14 | -69 |
| | | Sn | 0.15 | 0.21 | 0.07 | 0.31 | Sn | 0.14 | 0.31 | 0.07 | 0.27 |
| | $C_1(C_{12}(u_1, u_2; \theta_{12}), u_3; \theta_1)$ | $\theta_{12}$ | 1.37 | 0.9 | 3.51 | 1.41 | $\theta_{12}$ | 1.57 | 0.91 | 4.26 | 1.76 |
| | | $\theta_1$ | 1.59 | 0.93 | 4.75 | 1.73 | $\theta_1$ | 1.88 | 1.37 | 5.98 | 2.11 |
| | | AIC | -79.69 | -71.27 | **-102.84** | -54.29 | AIC | -99.7 | -79.76 | **-112.93** | -78.49 |
| | | Sn | 0.12 | 0.11 | **0.03** | 0.3 | Sn | 0.08 | 0.18 | **0.03** | 0.19 |
| Barley | $C(u_1, u_2, u_3; \theta)$ | $\theta$ | 1.43 | 0.66 | 3.25 | 1.57 | $\theta$ | 1.58 | 0.81 | 4.12 | 1.8 |
| | | AIC | -80.8 | -78.91 | -101.84 | -57.51 | AIC | -105.59 | -85.87 | -118.55 | -83.54 |
| | | Sn | 0.12 | 0.21 | 0.07 | 0.26 | Sn | 0.16 | 0.36 | 0.08 | 0.3 |
| | $C_1(C_{12}(u_1, u_2; \theta_{12}), u_3; \theta_1)$ | $\theta_{12}$ | 1.38 | 0.87 | 3.54 | 1.43 | $\theta_{12}$ | 1.72 | 1.05 | 5.04 | 1.99 |
| | | $\theta_1$ | 1.7 | 0.92 | 4.89 | 1.92 | $\theta_1$ | 1.94 | 1.41 | 6.02 | 2.21 |
| | | AIC | -95.8 | -72.07 | **-107.17** | -73.98 | AIC | -112.52 | -86.85 | -116.31 | -90.86 |
| | | Sn | 0.09 | 0.12 | **0.04** | 0.22 | Sn | 0.08 | 0.21 | **0.03** | 0.19 |

conditions aggravate from moderate to extreme stress, crop loss increases 6 to 8% (depending on the cereal and region). For comparison, conditional cumulative probability distributions for single stressors compared with the compound stressors are shown in Figure A.2 for all three severity levels.

While Figure 6 illustrates the same severity levels for the different hazards, Figure 7 illustrates crop loss for a range of different combinations of severity levels of dry and hot conditions (e.g., extreme dry conditions combined with moderate, severe and extreme hot conditions, and vice-versa) starting from the $50^{\text{th}}$ percentile of $P_{\text{MAM}}$ and $Tmax_{\text{MAM}}$. When $P_{\text{MAM}}/Tmax_{\text{MAM}}$ are below/above the median, the probability of crop loss is always higher than 40%. Similarly to Figure 6, the increase of crop loss with the severity of drought- and heat-stress is evident (Figure 7). The higher likelihood of crop loss in Cluster 2,

particularly for barley, is also consistent with Figure 6. Moreover, the results indicate that droughts are typically associated with higher probabilities of crop loss than heatwaves at the same severity level. This finding suggests that drought stress causes more damage to crop yields than heat stress, even for lower values of stress.

    In all cases, the additional effect of compound dry and hot conditions is larger when starting from only hot conditions, compared to when starting from only dry conditions (Figure 8 for moderate stress, Figure A.3a and b for severe and extreme

stress). The estimates are based on Equations 4 and 5. Depending on the cereal and region, the difference from drought stress to compound conditions may vary from 8% (barley in Cluster 1) to 11% (barley in Cluster 2). In contrast, the difference from



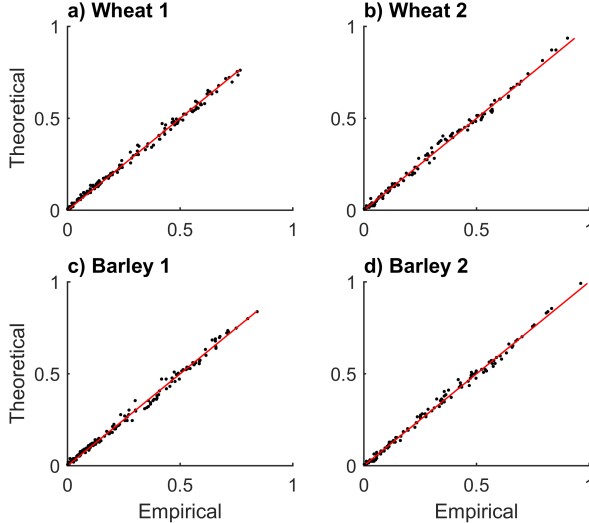

**Figure 4.** Empirical versus theoretical probability distributions based on the nested Frank copula models. (a) Wheat in Cluster 1. (b) Wheat in Cluster 2. (c) Barley in Cluster 1. (d) Barley in Cluster 2.

heat stress to compound conditions may vary between 19% (barley in Cluster 2) to 29% (wheat in Cluster 2). Uncertainties are large for these estimates and increase with the severity of the events (Figure A.3). Consistent with Figure 7 these findings suggest that drought stress is the major driver of crop loss associated with compound drought and heat.

## 4 Discussion

We have modelled the trivariate relationship between $Tmax_{MAM}$ and $P_{MAM}$ and wheat and barley yields in two regions in Spain using nested copulas. We found that the likelihood of crop loss increases with the severity of the compound dry and hot conditions and that compound drought and heat always increases the likelihood of crop loss. Moreover, our findings suggest that drought stress does not require to be so extreme as heat stress to cause the same adverse impact on crop yields. Hence drought is the more stressful driver of crop loss, when considering compound drought and heat.

Although the use of different methodologies, spatio-temporal scales and the focus on different cereals and regions makes a comparison between studies difficult, our findings are consistent with previous work. Using bivariate return periods of combined climate conditions, Zscheischler et al. (2017) have shown how linear models based directly on precipitation and temperature (and not the respective bivariate return period) may underestimate the explained variability of crop yields and that in several countries maize yields decrease with dry and hot conditions. Based on a meta-Gaussian model at the national level, Feng et al. (2019) have also shown that compound dry and hot extremes lead to larger impacts on maize yields than the individual hazards over five major maize-producing countries.

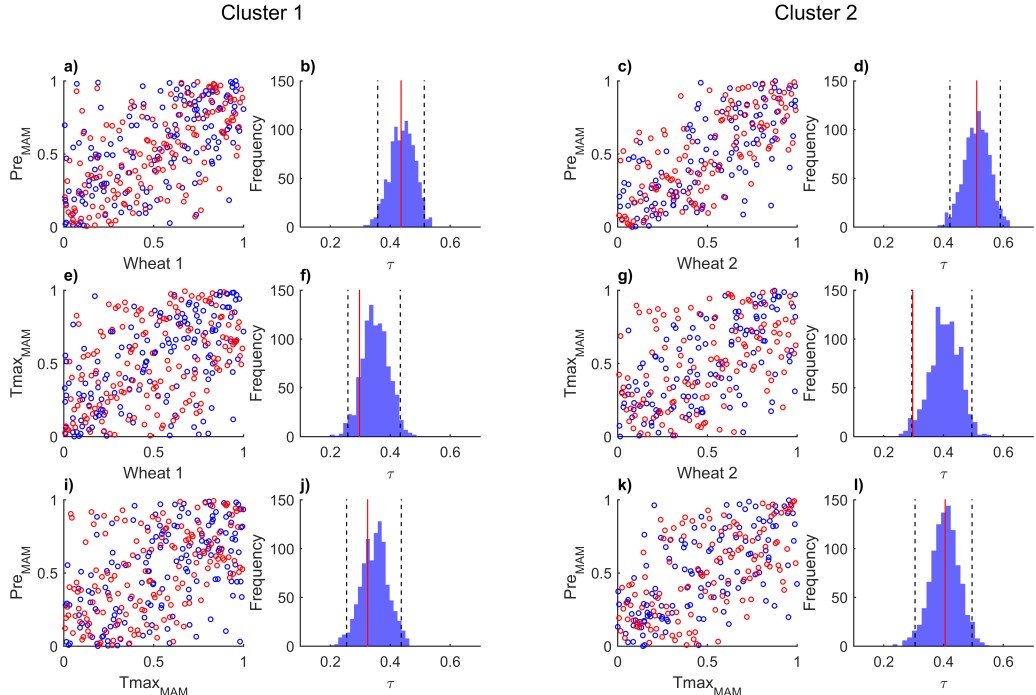

**Figure 5.** Scatterplots of copula-based samples (blue) compared with ranked observations (red) of crop anomalies with climate variables ($P_{MAM}$ and $Tmax_{MAM}$) (a), c), e) and g)) and $P_{MAM}$ against $Tmax_{MAM}$ (i and k)), for both clusters. The histograms (b), d), f), h), j), l)) correspond to the Kendall rank correlation of each pair based on 10,000 simulations with the same sample size of the observational sample. The 95% confidence intervals are shown with dashed lines. The red lines indicate the Kendall rank correlation of the observations.

In terms of the relative contributions of drought and heat conditions, a variety of studies at the national scale have found that the response varies from country to country. Feng et al. (2019) have found that China, France and Romania expect higher

chances of maize loss under dry conditions with normal temperatures (rather than under hot conditions with normal precipitation), while USA and Argentina expect higher chances of maize loss under hot conditions with normal precipitation (rather than under dry conditions with normal temperatures). In contrast, Zscheischler et al. (2017) have found that countries such as Lithuania, Luxembourg, and the UK, maize yields increase under hot and wet conditions, likely because of the importance of summer precipitation for the crop vegetative cycle and the relatively cooler climate in those countries.

Although previous studies have discussed that maximum temperature might be the best predictor variable for yield variability in most countries (Zscheischler et al., 2017), our study highlights that in Spain crop loss of wheat and barley is more sensitive to dryness than to hot conditions. This finding agrees with the rainfed practices adopted in the wheat and barley cultivation in Spain. In fact, the nesting structure of the trivariate models adopted in the present study privileges the stronger dependency between yields and precipitation, rather than between yields and temperature or between precipitation and temperature (Figure 3).





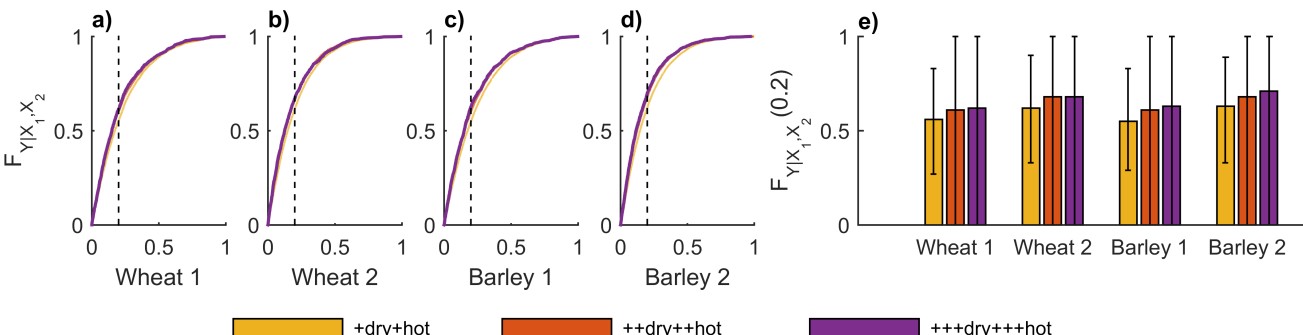

**Figure 6.** Conditional probability distributions of crop yield anomalies $F_{Y|X_2,X_2}$ over each cluster of provinces (wheat in Cluster 1 (a), wheat in Cluster 2 (b), barley in Cluster (1) and barley in Cluster 2 (d)) under moderate (+dry+hot, yellow, $P_{MAM}$ below the 20th and $Tmax_{MAM}$ above the 80th percentile), severe (++dry++hot, orange, $P_{MAM}$ below the 10th and $Tmax_{MAM}$ above the 90th percentile) and extreme (+++dry+++hot, purple, $P_{MAM}$ below the 5th and $Tmax_{MAM}$ above the 95th percentile) compound dry and hot conditions. (e) Conditional probabilities of non-exceeding the crop loss threshold (20th percentile – vertical dashed line in a-d)) for each severity level of compound dry and hot conditions given by $F_{Y|X_1,X_2}(0.2)$. Uncertainty ranges illustrate the 95% confidence intervals.

Though irrigated crops typically produce higher yields, the pressure in water resources is already increasing the deficit between water supplies and water demand in Spain (Rodríguez Díaz et al., 2007). Hence, understanding climate risks for rainfed crops is crucial to address the current water management challenges for agricultural practices in Mediterranean regions.

Higher probabilities of crop loss under drought and/or heat stress are generally expected in the southern region of Spain, in comparison to the northern region (Figures 6 and 7), in agreement with the higher temperatures and lower rainfall amounts

observed in the south (Ribeiro et al., 2019a; IM and AEMET, 2011). In the case of wheat losses, this finding is in agreement with previous work which focused on drought risks for the same crops and the same region (assessed based on remote sensing and hydro-meteorological drought indicators, Ribeiro et al. (2019b). However, Ribeiro et al. (2019b) identified a higher likelihood of barley loss with drought in the northern cluster. This discrepancy underlines importance of addressing the interaction between compound dry and hot conditions and the associated impacts on vegetation. For instance, compound dry and hot con-

ditions have a larger impact on the carbon uptake potential than the sum of the individual impacts (Zscheischler et al., 2014), highlighting the relevance of interactions between multiple stressors.

We found that for barley in Cluster 2, drought is the least dominant driver in comparison to the other cereals and regions. Barley in Cluster 2 shows the highest difference between drought and compound dry and hot conditions, and the lowest difference between heat stress and compound conditions (Figure 8). This suggests that among both cereals and both regions,

barley in Cluster 2 is the case where the compound and possibly interacting effects of drought and heat are most relevant. Note that in this case also the ECDF's between the dry and hot and dry or hot conditions are more differentiated from each other for the severe and extreme stress (Figure A.2). This is consistent with a recent study at the province level, which recommended that crop production in Spain should focus more on wheat production given that most provinces displayed lower levels of wheat loss with drought in comparison to barley loss (Ribeiro et al., 2019a). This finding is also consistent with Figures 6 and 7.


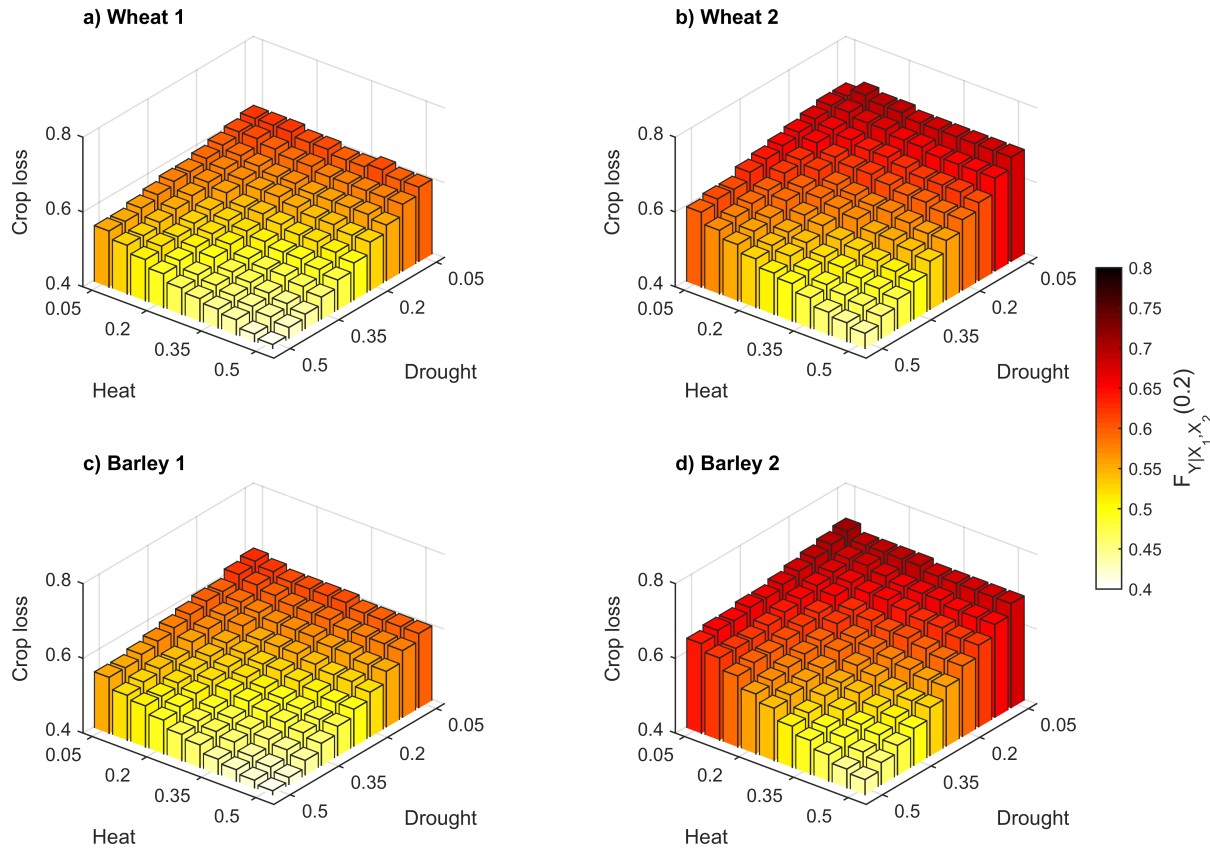

**Figure 7.** Conditional probability of crop loss given by $F_{Y|X_1,X_2}(0.2)$ (bar height) for both clusters and cereals (wheat in Cluster 1 (a), wheat in Cluster 2 (b), barley in Cluster (1) and barley in Cluster 2 (d)) for different combinations of severity levels of dry and hot conditions. The x-axis indicates the $P_{MAM}$ percentiles (Drought) and y-axis indicates the $Tmax_{MAM}$ percentile (Heat).

The uncertainties associated to the parametric statistical model were assessed with a large number of sampled distributions with the same sample size as the observations. In some of these distributions, drought or heat alone may cause more damage than concurrent drought and heat (lower uncertainty bound is below 0 in Figures 8 and A.3). This highlights the challenges of estimating the likelihood of rare events in two- or three-dimensional probability distribution with limited sample size (Serinaldi, 2013, 2016; Zscheischler and Fischer, in review). For the same reason, the wheat loss in Cluster 2 when $P_{MAM}$ is below the $5^{th}$
percentile in Figure 7 slightly decreases when the threshold of $Tmax_{MAM}$ change from the $10^{th}$ percentile to the $5^{th}$ percentile (while an increase would be expected like in the other cases). Note that the uncertainties increase with the increasing severity of the compound dry and hot conditions (Figure 6 and 8) due the rapid decrease of available samples in the corners of the three-dimensional probability distribution. Moreover, following the work by Okhrin and Ristig (2014), here we considered





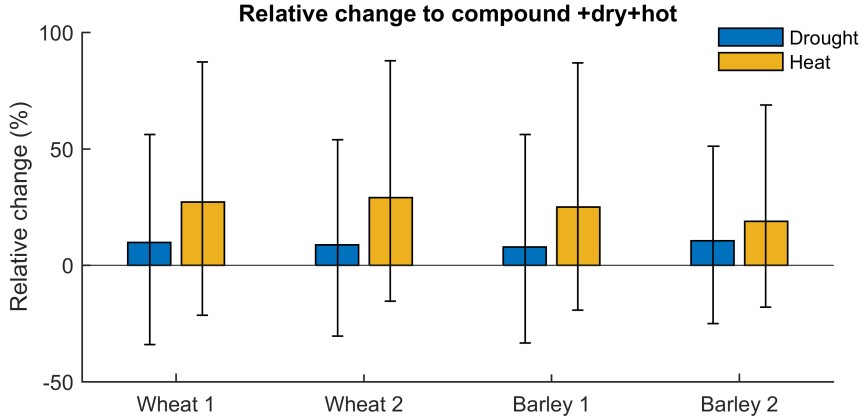

**Figure 8.** Difference in probability of crop loss from dry (blue) and hot (orange) to compound dry and hot conditions in wheat (left) and barley (right) for Cluster 1 and 2. Shown are the best estimates for moderate dry and hot (+dry+hot) conditions (bar height) and associated 95% confidence intervals.

nesting copulas of the same family only, as more complex structures would be difficult to implement in general. Vine copulas

might offer an alternative that is also appropriate for higher dimensions (Bevacqua et al., 2017), when considering for instance more driver variables. Nevertheless, in comparison with previous studies based on bivariate models only, we argue that the statistical modelling based on NAC is a good compromise between complexity and the trivariate dimension.

## 5   Conclusions

The present study assessed how compound drought and heat enhance losses of wheat and barley in two major dryland areas

in Spain. We showed that nested Archimedian copulas can successfully model the trivariate joint distribution between spring maximum temperature, spring precipitation and yields to estimate conditional probabilities of crop loss under different severity levels of hot and dry conditions. The strongest dependence exists between spring precipitation and yields and is best captured by a Frank copula. Our results demonstrate that the probability of crop loss increases with the severity of compound dry and hot conditions. Furthermore, the likelihood of wheat and barley loss increases when drought or heat, respectively, aggravate

to compound dry and hot conditions in both regions. Overall, the likelihood of crop loss in the southern region is larger, in particularly for barley. For both cereals and regions, the likelihood of crop loss increase more with increasing drought stress than with heat stress, suggesting that drought plays a dominant role in the compound event. Our results illustrate the additional value of using trivariate copula modelling to estimate the compounding effects of dry and hot extremes on the risk of crop failure. In operational practice, this research will allow contributing to design supporting tools and provide guidance in the

decision-making process in agricultural practices to minimize crop losses related to climate hazards.

*Code and data availability.* The statistical analysis was performed under R software using the packages *copula* and *HAC*. The precipitation- and maximum temperature gridded values are publicly available from the Climate Research Unit (CRU) TS4.01 dataset by Harris et al. (2014). The Spanish crop yield is published by the Spanish Agriculture, Fishing and Environment Ministry in their Statistical Yearbooks, which can be consulted at https://www.mapa.gob.es/es/estadistica/temas/publicaciones/anuario-de-estadistica/ (last access on 9 November 2019). The CORINE Land Cover datasets are publicly available at https://land.copernicus.eu/pan-european/corine-land-cover.


*Author contributions.* A. F. S. Ribeiro analyzed the data, produced the figures and drafted the manuscript. J.Zscheischler supervised the overall work with an emphasis on the design of the statistical framework. A. Russo and C. M. Gouveia helped to supervise the work and conceived the original idea together with A.F.S.Ribeiro. P. Páscoa instructed the acquisition and analysis of the crop yield data. All the authors discussed the results, provided critical feedback and helped shape the research, analysis and contributed to the final manuscript.

*Competing interests.* The authors declare that they have no conflict of interest.

*Acknowledgements.* A.F.S. Ribeiro is thankful to the COST Action CA17109 for a Short Term Scientific Mission (STSM) grant to develop the present work and to the Climate and Environmental Physics Department of the University of Bern which was the host institution. This work was also partially supported by Portuguese funds through FCT (Fundação para a Ciência e a Tecnologia, Portugal) under the projects CLMALERT (ERA4CS/0005/2016) and IMPECAF (PTDC/CTA-CLI/28902/2017). A.F.S. Ribeiro also thanks FCT for grant 295  PD/BD/114481/2016. J. Zscheischler acknowledges funding from the Swiss National Science Foundation (Ambizione grant 179876).

**Appendix A: Supplementary material**





**Table A.1.** Kendalls' correlation ($\tau$) between crop yield ($u_1$), $P_{MAM}$ ($u_2$) and $Tmax_{MAM}$ ($u_3$). Maximum values are denoted in boldfor each cereal and cluster indicating the pair of variables with the strongest relationship. Bivariate copulas parameter ($\theta$), Akaike Information Criteria (AIC) and Cramer-von Mises distance (Sn) to respective empirical copula, considering the possible pairs of variables. Fit based on maximum pseudo-likelihood (Gumbel (G), Clayton (C), Frank (F) and Joe (J) copulas). Smallest values of AIC and Sn indicate the selected copula for each pair (bold).

| | | | Cluster 1 | | | | | Cluster 2 | | | | |
|---|---|---|---|---|---|---|---|---|---|---|---|---|
| | | | $\tau$ | G | C | F | J | $\tau$ | G | C | F | J |
| | | $\theta$ | | 1.59 | 0.93 | 4.75 | 1.73 | | 1.88 | 1.37 | 5.98 | 2.11 |
| | $C(u_1,u_2)$ | AIC | **0.44** | -51.43 | -47.28 | **-69.71** | -35.58 | **0.51** | -71.04 | -64.6 | **-81.22** | -53.26 |
| | | Sn | | 0.06 | 0.14 | **0.01** | 0.17 | | 0.04 | 0.11 | **0.02** | 0.13 |
| Wheat | | $\theta$ | | 1.28 | 0.71 | 2.73 | 1.27 | | 1.31 | 0.53 | 2.88 | 1.38 |
| | $C(u_1,u_3)$ | AIC | 0.30 | -14.3 | -31.71 | -28.51 | -4.07 | 0.30 | -13.83 | -13.07 | -23.77 | -8.08 |
| | | Sn | | 0.09 | 0.04 | 0.03 | 0.18 | | 0.08 | 0.1 | 0.03 | 0.13 |
| | | $\theta$ | | 1.4 | 0.58 | 3.27 | 1.51 | | 1.66 | 0.77 | 4.28 | 1.98 |
| | $C(u_2,u_3)$ | AIC | 0.32 | -28.45 | -21.74 | -38.13 | -20.41 | 0.41 | -52.05 | -27.27 | -48.85 | -47.27 |
| | | Sn | | 0.07 | 0.11 | 0.03 | 0.13 | | 0.04 | 0.14 | 0.03 | 0.08 |
| | | $\theta$ | | 1.7 | 0.92 | 4.89 | 1.92 | | 1.94 | 1.41 | 6.02 | 2.21 |
| | $C(u_1,u_2)$ | AIC | **0.44** | -66.25 | -47.07 | **-72.18** | -53.18 | **0.51** | -78.79 | -68.34 | **-81.99** | -61.18 |
| | | Sn | | 0.02 | 0.13 | **0.02** | 0.08 | | 0.03 | 0.1 | **0.02** | 0.1 |
| Barley | | $\theta$ | | 1.3 | 0.69 | 2.77 | 1.31 | | 1.46 | 0.69 | 3.73 | 1.61 |
| | $C(u_1,u_3)$ | AIC | 0.30 | -16.34 | -30.27 | -29.9 | -6.11 | 0.38 | -29.33 | -22.43 | -38.56 | -21.43 |
| | | Sn | | 0.08 | 0.06 | 0.04 | 0.16 | | 0.09 | 0.15 | 0.04 | 0.16 |
| | | $\theta$ | | 1.4 | 0.58 | 3.27 | 1.51 | | 1.66 | 0.77 | 4.28 | 1.98 |
| | $C(u_2,u_3)$ | AIC | 0.32 | -28.45 | -21.74 | -38.13 | -20.41 | 0.41 | -52.05 | -27.27 | -48.85 | -47.27 |
| | | Sn | | 0.07 | 0.11 | 0.03 | 0.13 | | 0.04 | 0.14 | 0.03 | 0.08 |

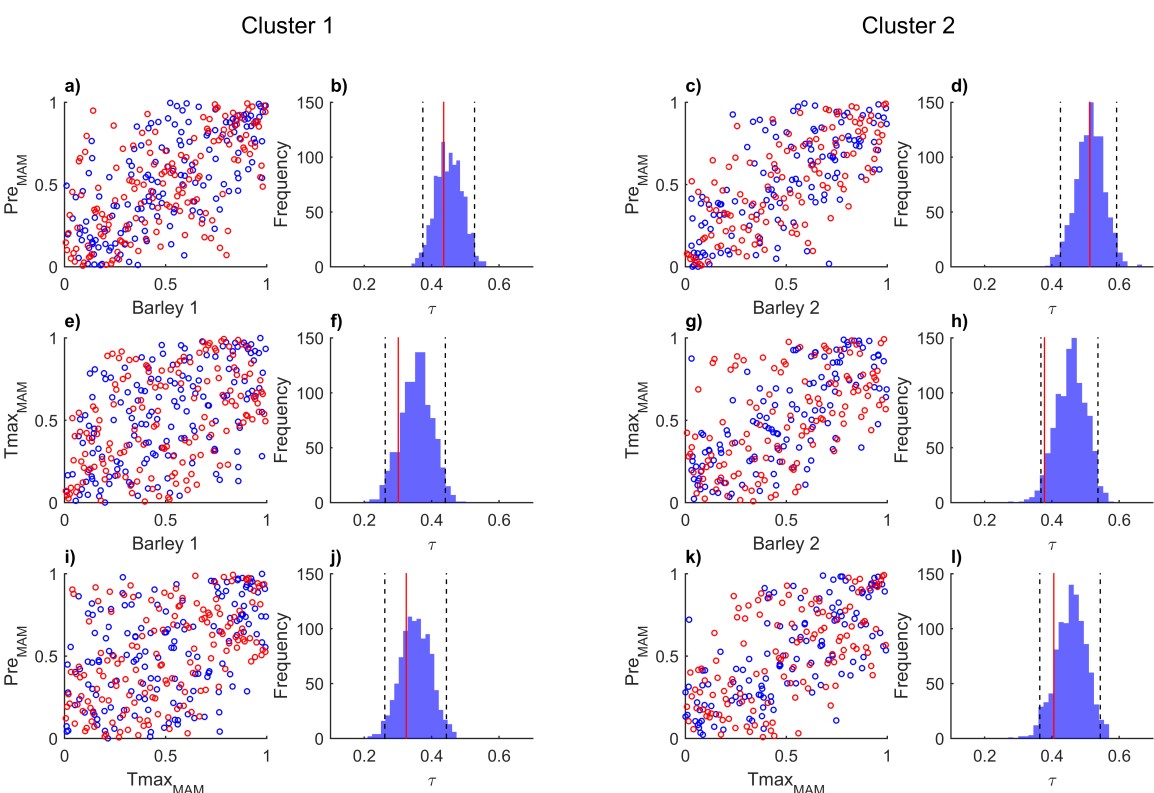

**Figure A.1.** Same as Figure 5 but for barley.





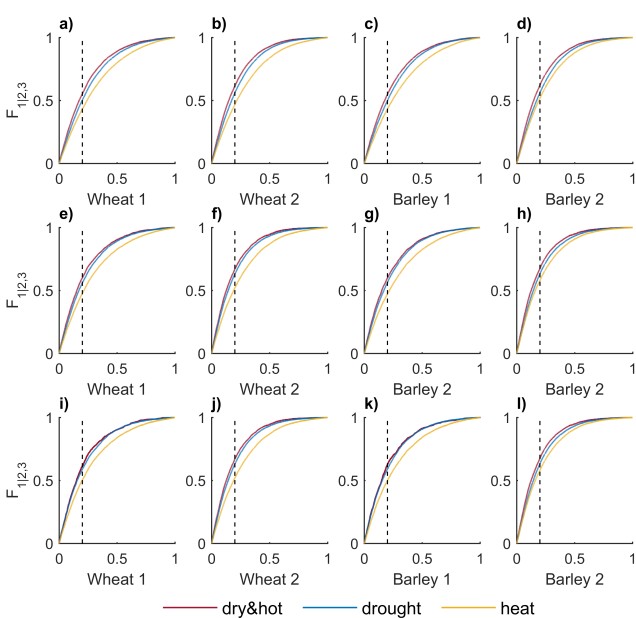

**Figure A.2.** Conditional probability distributions of crop yield anomalies over each cluster under hot (yellow), dry (blue) or compound dry and hot (purple) under moderate (a) - d)), severe (e) - h)) and extreme conditions (i) - l)).

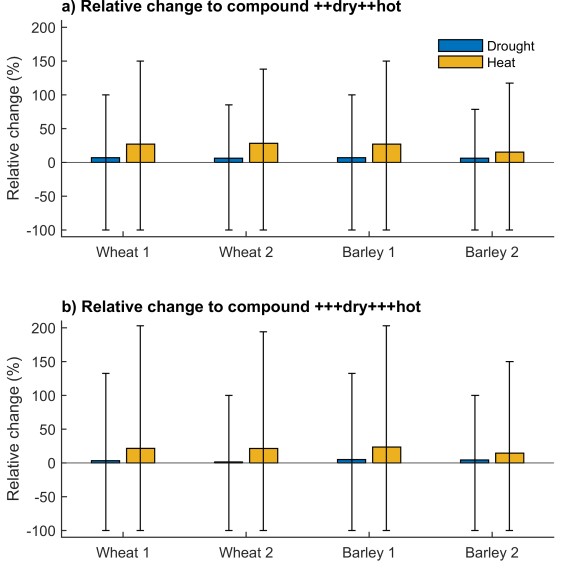

**Figure A.3.** Same as Figure 8 but for severe (a) and extreme (b) conditions.



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
