# Peer review of "Risk of crop failure due to compound dry and hot extremes estimated with nested copulas"

_Biogeosciences, 2020_

## Referee Comment (RC1) · Anonymous Referee #1 · 14 Apr 2020

This study evaluated the risk of crop failure due to compound dry and hot extremes. A copula model is fitted to estimate the response of crop yield with respect to different dry and hot conditions. This manuscript is well crafted with clear structure. A few issues need to be addressed before the potential publication of this study.

(1) Selection of the periods Line 92-93: "We used 3-monthly means of Tmax and 3-monthly means of P during spring". Here the selection is based on the correlation analysis, but not the whole growing season, right? Please justify this period. It is easy to understand this from a statistical perspective. Is this selection still valid from a physical perspective?

(2) Copula implementation Line 161: "Due to the negative dependence between Tmax-MAM and both crop yields", The clayton copula does not permit the negative dependence. Is this the reason to "invert the margins of TmaxMAM for copula modelling"? The rationale of this transformation needs to be clarified. Suggest to make it clear to aid the understanding.

(3) Figure presentation Figure 7: "y-axis indicates the TmaxMAM percentile (Heat)". For heat, should you use the axis with the range like 0.5-0.95? Since for heat, we are interested in high percentile, right? Or if this is related to the aforementioned "inversion of the margins", please clarity this and make it clear.

(4) Figure discussion Regarding Figure 7, "When PMAM/TmaxMAM are below/above the median, the probability of crop loss is always higher than 40%." How could you tell this (i.e., above the median?) from the figure? The y-axis for heat stress is below median. Please make it clear.

Minor comments: Check the bracket in the caption of Figure 5.

---

## Referee Comment (RC2) · Anonymous Referee #2 · 20 Apr 2020

The manuscript 'Risk of crop failure due to compound dry and hot extremes estimated with nested copulas' uses Archimedian copulas to model trivariate joint distributions between maximum temperature, precipitation and wheat and barley yield deviations. The paper is well structured and well written. It contributes interesting new insights to the literature on compound dry and heat impacts on crop yields.

A few comments include:

p.4, l.88: I recommend adding the exact months rather than just naming the seasons. As readers of the paper might come from all over the world, it might not be clear which months include the spring time in Spain. p.6, l.132: The copula parameter $\theta$ should be

introduced

p.8, l.178: Can you interpret what it means that 'C_(u1,u2,u3) is slightly lower than the AIC of C_1 (u3,C_12 (u1,u2))' in terms of compound and single hazards and add a sentence about it?

*Code availability. It is more and more common to publish the code used in scientific publications and I strongly recommend the authors considering to publish their code once the paper is accepted.

---

## Referee Comment (RC3) · Anonymous Referee #3 · 5 Jun 2020

This is a well-written manuscript that investigates the compound effects of precipitation and temperature on crop failure in two provinces in Spain using nested copulas. It contributes to a better understanding of these type of compound events and it therefore deserves publication.

I have just some minor comments that I recommend the authors to address before publication:

p.4 line 90. Is it monthly daily mean precipitation or monthly accumulated precipitation? Please specify.

p.5 Figure 2. I would highlight the month MAM as it is the choice to of this study to

perform the compound event analysis.

p 7. line 152. I think this Section should explain what the choice for the marginals is and why.

p 7. line 162 The authors say they invert the margins due to negative dependence between temperature and precipitation. Why not use a rotated copula that can represent negative correlation instead?

p.8 line 183 For wheat 2, it seems that the statistical model tend to produce a larger kendal correlation between temperature and wheat (FIgure 5g) than what is seen in the observations (the observations are almost outside the confident interval obtained from simulations). Could the authors explain why the performance for this specific case seems to be worse?

p. 13 line 256 The authors say that in some cases, draught or heat alone may cause more damage than concurrent drought and heat. I see this is the case for wheat 2 (Figure 7b). Any physical explanation to this? I would have assumed that regardless of draught playing a greater role, extreme values of these variables would both contribute to increase yield loss.

---

## Editor Comment (EC1) · Andreia Filipa Silva Ribeiro et al. · 18 Jun 2020

The three reviewers agree on the clear purpose and structure of the manuscript and are all supportive to publication of this manuscript subject to minor edits. Some refer to justification of (implicit) choices made, some ask for some extra clarification. I would recommend to reply to all review comments that are raised. For me the most significant comments are:

Additional justification regarding:

- The choice of the 3-month averaging window for the meteorological quantities

[Figure]

- The processing of the negative dependence (why not use a rotated copula, as suggested by a reviewer)

- Availability of the code (and data if that is open)

Some further explanation concerning:

- The copula parameter {theta

- The implication of C being lower than the AIC

- The choice of the marginals in section 2.3

- The reconstruction of the correlation between yield and temperature for wheat 2

- The physical interpretation, not only of the notion that drought and heat alone can give stronger effects than their combination, but in general sense: why would the combination of environmental drivers lead to stronger yield reductions? Is this physiologically explainable?

Some editing would be recommended regarding:

- The renaming of "cluster" to "region" (as "cluster" does not have a strong geographical association)

- The labeling of heat percentiles in Figure 7.

With kind regards

---

## Author Comment (AC1) · 24 Jun 2020

Reviewer: This study evaluated the risk of crop failure due to compound dry and hot extremes. A copula model is fitted to estimate the response of crop yield with respect to different dry and hot conditions. This manuscript is well crafted with clear structure.

Author's Reply: Thank you for this positive assessment.

Reviewer: A few issues need to be addressed before the potential publication of this study. (1) Selection of the periods Line 92-93: "We used 3-monthly means of Tmax and 3- monthly means of P during spring". Here the selection is based on the correlation

analysis, but not the whole growing season, right? Please justify this period. It is easy to understand this from a statistical perspective. Is this selection still valid from a physical perspective?

Author's Reply (1): Thank you for the question. Given the importance of assessing crop's water and temperature requirements at different moments of the vegetative cycle we conducted a correlation analysis between the yield and the 3-monthly means of precipitation and 3-monthly means of maximum temperature during the whole growing season (approximately from September of year n-1 to June of the year n), as shown in Fig. 2. The identification of the moment of the vegetative cycle of the highest crop's water and temperature requirements was assessed based on the strongest statistically significant correlation value. Fig. 2 suggests that the greatest influence of P and Tmax in crop yields is observed during spring (in both regions and cereals) corresponding to the moments in which the vegetation is photosynthetically more active. The effects of water content and high temperatures during middle growth stages of the crop life cycle are in accordance with previous studies (Ferrise et al., 2011; García del Moral et al., 2003; Iglesias and Quiroga, 2007; Ribeiro et al., 2019). Hence, this selection is valid both from the statistical and biophysical point of view.

We will clarify this aspect in the revised manuscript to the following in the Weather data section: "The vegetative cycle of the winter crops in Spain is mainly driven by precipitation and temperature: sowing occurs around autumn, followed by the vegetative phase in winter, reproductive phase (more photosynthetically active phase) in spring and crop harvest occurs in the early summer. Therefore, monthly precipitation (P) and monthly maximum temperature (Tmax) were extracted from the Climate Research Unit (CRU) TS4.01 dataset (Harris et al., 2014) spanning the same time period. Given the importance of assessing crop's water and temperature requirements at different moments of the vegetative cycle we conducted a correlation analysis between the annual yields and the 3-monthly means of P and 3-monthly means of Tmax during the whole growing season, as shown in Fig. 2. The identification of the moment of the vegetative cycle

of the highest crop's water and temperature requirements was assessed based on the strongest statistically significant correlation value (denoted by filled circles in Fig. 2). Figure 2 suggests that the greatest influence of P and Tmax in crop yields is observed during spring (MAM in both regions and cereals) corresponding to the reproductive phase of plant development, when vegetation is photosynthetically more active. In this way, we used 3-monthly means of Tmax and 3-monthly means of P during spring (PMAM and TmaxMAM, respectively), which has also been identified in previous studies as a growth stage sensitive to the effects of water content and high temperatures (Ferrise et al., 2011; García del Moral et al., 2003; Iglesias and Quiroga, 2007; Ribeiro et al., 2019). This selection of climate variables allows to maximize the dependence between climate conditions and yields as also shown by previous work based on the same data (Ribeiro et al., 2019c)."

References:

Ferrise, R., Moriondo, M. and Bindi, M.: Probabilistic assessments of climate change impacts on durum wheat in the Mediterranean region, Nat. Hazards Earth Syst. Sci., 11(5), 1293–1302, doi:10.5194/nhess-11-1293-2011, 2011.

García del Moral, L. F., Rharrabti, Y., Villegas, D. and Royo, C.: Evaluation of Grain Yield and Its Components in Durum Wheat under Mediterranean Conditions: An Ontogenic Approach, Agron. J., 95, 266–274, 2003.

Iglesias, A. and Quiroga, S.: Measuring the risk of climate variability to cereal production at five sites in Spain, Clim. Res., 34(1), 47–57, doi:10.3354/cr034047, 2007.

Reviewer: (2) Copula implementation Line 161: "Due to the negative dependence between TmaxMAM and both crop yields", The clayton copula does not permit the negative dependence. Is this the reason to "invert the margins of TmaxMAM for copula modelling"? The rationale of this transformation needs to be clarified. Suggest to make it clear to aid the understanding.

Author's Reply (2): Thank you for the comment. The reason for inverting the margins is that the required complete monotonicity of the ACs generators to construct NAC following Okhrin and Ristig (2014) implies (i) that the same single-parameter generator function is used on each level of NAC (i.e. same family), but potentially with a different value of $\theta$ (as we discuss in lines 53-56, 135-40 and 263-264 in other words) and (ii) positively dependent AC models, hence the pairwise rank correlations are required to be non-negative. Therefore, in order to model positive dependencies among all possible pairs, we considered the inverted values of Tmax (i.e. multiplication by $-1$). For more details on complete monotonicity of the ACs generators and NAC constructions see e.g. Górecki et al. (2017).

We will clarify this in the revised manuscript by moving the referred information in line 161 (as the required complete monotonicity of the AC generators implies both conditions) and improving to: "Using the same single-parameter generator function on each level of NAC (but with a potentially different value of $\theta$) satisfies the required complete monotonicity of the ACs generators to construct NAC following Okhrin and Ristig (2014), which also implies that the possible pairs are positively dependent. Therefore, due to the negative dependence between TmaxMAM and both crop yields and PMAM, we inverted the margins of TmaxMAM for copula modelling (i.e. multiplication by $-1$). For more details on complete monotonicity of the ACs generators and NAC constructions see e.g. Górecki et al. (2017)."

References:

Górecki, J., Hofert, M. and Holeňa, M.: On structure, family and parameter estimation of hierarchical Archimedean copulas, J. Stat. Comput. Simul., 87(17), 3261–3324, doi:10.1080/00949655.2017.1365148, 2017.

Reviewer: (3) Figure presentation Figure 7: "y-axis indicates the TmaxMAM percentile (Heat)". For heat, should you use the axis with the range like 0.5-0.95? Since for heat, we are interested in high percentile, right? Or if this is related to the aforementioned

"inversion of the margins", please clarity this and make it clear.

Author's Reply (3): You are correct. By inverting the Tmax the highest values correspond to the lower quantiles. We will change the Figure y-axis to 0.5 - 0.95 to avoid confusion.

Reviewer: (4) Figure discussion Regarding Figure 7, "When PMAM/TmaxMAM are below/above the median, the probability of crop loss is always higher than 40%." How could you tell this (i.e., above the median?) from the figure? The y-axis for heat stress is below median. Please make it clear.

Author's Reply (4): We agree that this point is not clear. In the text we refer to the Tmax values, rather than the inverted Tmax values as it was supposed to. In other words, when referring to Tmax in the text we keep the concept of exceeding the highest percentiles. As mentioned above in comment (3), we will change the Figure y-axis to 0.5 - 0.95 to avoid this confusion.

Reviewer: (5) Minor comments: Check the bracket in the caption of Figure 5.

Author's Reply (5): Thank you, we will delete the extra brackets in Fig. 5 caption

[Figure]

**Fig. 1.** Updated Figure 7

---

## Author Comment (AC2) · 24 Jun 2020

Reviewer: The manuscript 'Risk of crop failure due to compound dry and hot extremes estimated with nested copulas' uses Archimedian copulas to model trivariate joint distributions between maximum temperature, precipitation and wheat and barley yield deviations. The paper is well structured and well written. It contributes interesting new insights to the literature on compound dry and heat impacts on crop yields.

Author's Reply: Thank you for this positive assessment.

Reviewer: A few comments include: (1) p.4, l.88: I recommend adding the exact

months rather than just naming the seasons. As readers of the paper might come from all over the world, it might not be clear which months include the spring time in Spain.

Author's Reply (1): Thank you for the comment. To clarify the season months in the study region we propose to rewrite the lines 88 – 90 to the following:

"The vegetative cycle of the winter crops in Spain is mainly driven by precipitation and temperature: sowing occurs around autumn (from September through November, SON), followed by the vegetative phase in winter (from December through January, DJF), reproductive phase (more photosynthetically active phase) in spring (from March through May, MAM) and crop harvest occurs in the early summer (around June)"

Reviewer: (2) p.6, l.132: The copula parameter $\theta$ should be introduced

Author's Reply (2): Thank you for the suggestion. We propose to rewrite the lines 127-128 to:

"AC can be written in terms of the respective generator function $\varphi$, which belongs to a parametric family $(\varphi\_\theta)$ dependent on the parameter $\theta$, e.g. for the three-dimensional case:"

in lines 130-131 to:

"Due to the symmetry of bivariate AC, the above trivariate form can be expressed in terms of NAC or HAC, where two of the margins are first coupled by their bivariate copula and then coupled with the third margin, via the same generator on each level but different parameters $\theta\_12$ and $\theta\_1$, respectively, e.g.:"

And in lines 131-132 to:

Equation 8 can also be expressed in terms of the other possible pair copulas $C\_13$ $(u\_1,u\_3;\theta\_13)$ and $C\_23$ $(u\_2,u\_3;\theta\_23)$ that are coupled with $u\_2$ and $u\_1$ by $C\_2$ and $C\_3$, with expressions $C\_2 (C\_13 (u\_1,u\_3;\theta\_13),u\_2;\theta\_2)$ and $C\_3 (C\_23$

($u\_2,u\_3;\theta\_23$ ),$u\_1;\theta\_3$), respectively. Like Eq. (8), among each structure of NAC the same generator is required for each level but with different parameter, hence, both the optimal structure and respective parameters must be determined.

Reviewer: (3) p.8, l.178: Can you interpret what it means that 'C_(u1,u2,u3) is slightly lower than the AIC of C_1 (u3,C_12 (u1,u2))' in terms of compound and single hazards and add a sentence about it?

Author's Reply (3): This means that in the case of barley in Cluster 2, the trivariate copula fits the data slightly better than the two-parameter NAC C(u3, C12(u1,u2)) in terms of AIC, even though the Cramer-von Mises distance is better for the NAC. This could mean that in this case a NAC structure favouring the dependence between yield and precipitation may be less relevant compared to the other clusters and yields. Drought individually seems to play a less dominant role in the compound event, in comparison to the other cereals and regions.

This interpretation would be consistent with our discussion about Figure 8, where barley in Cluster 2 is also the case with the highest difference between drought and compound dry and hot conditions, hence illustrating that here drought is the least dominant driver of crop loss in comparison to the other cereals and regions.

Following the reviewer's suggestion we will add a sentence in the Results section: "The only exception is barley in Cluster 2 whose AIC of $C\theta$(u1,u2,u3) is slightly lower than the AIC of $C\theta1$ (u3,$C\theta12$ (u1,u2)) (Table 2). This feature may suggest that a structure favouring the dependence between yield and precipitation (u1,u2) may not be as relevant as in the other clusters and yields due to a less dominant role of drought individually in this case. Nevertheless, in terms of Cramer-von Mises distance (Sn) the nested copula is the closer to the empirical trivariate copula. For this reason, we modelled the trivariate joint distribution based on nested Frank copulas for all cases. (. . .)"

Reviewer: (4) *Code availability. It is more and more common to publish the code used

in scientific publications and I strongly recommend the authors considering to publish their code once the paper is accepted.

Author's Reply (4): We agree to publish the code in a repository.

---

## Author Comment (AC3) · 24 Jun 2020

Reviewer: This is a well-written manuscript that investigates the compound effects of precipitation and temperature on crop failure in two provinces in Spain using nested copulas. It contributes to a better understanding of these type of compound events and it therefore deserves publication.

Author's Reply: Thank you for this positive assessment.

Reviewer: I have just some minor comments that I recommend the authors to address before publication: (1) p.4 line 90. Is it monthly daily mean precipitation or monthly

accumulated precipitation? Please specify.

Author's Reply (1): Thanks for recommending the clarification, CRU TS4.01 provides monthly cumulative precipitation, hence we will clarify in the revised manuscript:

"Therefore, monthly accumulated precipitation (P) and monthly maximum temperature (Tmax) were extracted from the Climate Research Unit (CRU) TS4.01 dataset (...)"

Reviewer: (2) p.5 Figure 2. I would highlight the month MAM as it is the choice to of this study to perform the compound event analysis.

Author's Reply (2): We appreciate the suggestion and propose to denote MAM in bold text (please see updated Figure 2 further below).

Reviewer: (3) p 7. line 152. I think this Section should explain what the choice for the marginals is and why.

Author's Reply (3): Thank you for the comment. We used empirical ranks as explained in the methods sections. We suggest to add the following text:

The main steps of the trivariate approach used in this study can be summarized as follows (Okhrin and Ristig, 2014). First, the marginal distributions u1, u2 and u3 are estimated non-parametrically by simple ranking, using the empirical distribution functions of the data through the pobs R function, a common approach for copula modelling."

Reviewer: (4) p 7. line 162 The authors say they invert the margins due to negative dependence between temperature and precipitation. Why not use a rotated copula that can represent negative correlation instead?

Author's Reply (4): Thank you for the question. A similar comment was raised by the reviewer 1. As we also answered to reviewer 1, the required complete monotonicity of the ACs generators to construct NAC following Okhrin and Ristig (2014) implies (i) that the same single-parameter generator function is used on each level of NAC (i.e. same family), but with a different value of $\theta$ (as we discuss in lines 53-56, 135-40 and 263-

264 by other words) and (ii) positively dependent AC models, hence the pairwise rank correlations are required to be non-negative. Therefore, nested rotated copulas are not covered by the NAC approach following Okhrin and Ristig (2014). For this reason, in order to model positive dependencies among all possible pairs, we considered the inverted values of Tmax (i.e. multiplication by $-1$). For more details on complete monotonicity of the ACs generators and NAC constructions see e.g. Górecki et al. (2017).

We will clarify this in the revised manuscript by moving the referred information (as the required complete monotonicity of the AC generators implies both conditions) and improving to:

"Using the same single-parameter generator function on each level of NAC (but with a different value of $\theta$) satisfies the required complete monotonicity of the ACs generators to construct NAC following Okhrin and Ristig (2014), which also implies that the possible pairs are positively dependent. Therefore, due to the negative dependence between TmaxMAM and both crop yields and PMAM, we inverted the margins of TmaxMAM for copula modelling (i.e. multiplication by $-1$). For more details on complete monotonicity of the ACs generators and NAC constructions see e.g. Górecki et al. (2017)."

References: Górecki, J., Hofert, M. and Holeňa, M.: On structure, family and parameter estimation of hierarchical Archimedean copulas, J. Stat. Comput. Simul., 87(17), 3261–3324, doi:10.1080/00949655.2017.1365148, 2017.

Reviewer: (5) p.8 line 183 For wheat 2, it seems that the statistical model tend to produce a larger kendal correlation between temperature and wheat (FIgure 5g) than what is seen in the observations (the observations are almost outside the confident interval obtained from simulations). Could the authors explain why the performance for this specific case seems to be worse?

Author's Reply (5): As a matter of fact, a similar feature occurs in the case of barley

(Fig. A.1 - h). This applies for the correlation between Tmax and the crop yields. The explanation for this feature may be related to the construction of the NAC models, which is defined by the pair (P,yield) in the inner level due to their stronger correlation (Table A.1). In addition, the correlation between Tmax and yield, is also lower than the correlation between P and Tmax (Table A.1). For this reason, Tmax and yield is the pair with lowest correlation and hence the model is likely to struggle in its representation. Nevertheless, in both cases (wheat and barley), the simulated level of dependence is inside the 95% confidence level and the magnitude of correlations among the pairs is also preserved i.e., such that $\tau_{(u\_1,u\_2)} > \tau_{(u\_2,u\_3)} > \tau_{(u\_1,u\_3)}$.

We will clarify this on Results section of the revised manuscript:

"Bivariate dependencies as measured by Kendall's are captured well by the fitted models (Figure 5 for wheat, Figure A.1 for barley). Among all possible pairs, the correlation between Tmax and yield is the lowest for the case of both cereals (Table A.1), and for this reason it is the pair in Figure 5 and Figure A.1 with observational $\tau$ closest to the lower bound of the 95% confidence intervals (Figure 5f,h and Figure A.1f,h). Nevertheless, in both Figure 5 and Figure A.1, the simulated level of dependence is inside the 95% confidence level and the magnitude of correlations among the pairs is also reasonably preserved by the models i.e., such that $\tau_{(u\_1,u\_2)} > \tau_{(u\_2,u\_3)} > \tau_{(u\_1,u\_3)}$."

Reviewer: (6) p. 13 line 256 The authors say that in some cases, draught or heat alone may cause more damage than concurrent drought and heat. I see this is the case for wheat 2 (Figure 7b). Any physical explanation to this? I would have assumed that regardless of draught playing a greater role, extreme values of these variables would both contribute to increase yield loss.

Author's Reply (6): Thank you for the question. The best estimates (bars in Figures 8 and A.3) show indeed that compound dry and hot extremes contribute to increase yield loss. Nevertheless, the lower bound of the 95% confidence intervals in Figures 8 and

A.3 show that drought or heat alone may cause more damage than concurrent drought and heat due to uncertainties associated to the parametric statistical model. This is associated with the uncertainties in the estimation procedure, which may be particularly large for extreme values and it would be difficult to find a physical explanation for such a feature.

We will clarify this in the Discussion section in respect to uncertainties:

"The uncertainties associated to the parametric statistical model were assessed with a large number of sampled distributions with the same sample size as the observations. In some of these distributions, drought or heat alone may cause more damage than concurrent drought and heat (lower uncertainty bound is below 0 in Figures 8 and A.3). This highlights the challenges of estimating the likelihood of rare events in two- or three-dimensional probability distribution with limited sample size (Serinaldi, 2013, 2016; Zscheischler and Fischer, in review). For the same reason, the wheat loss in Cluster 2 when PMAM is below the 5th percentile in Figure 7 slightly decreases when the threshold of TmaxMAM change from the 10th percentile to the 5th percentile (while an increase would be expected like in the other cases). These features are associated with the uncertainties in the estimation procedure, which may be particularly large for extreme values and it would be difficult to find a physical explanation for such a feature. Note that the uncertainties increase with the increasing severity of the compound dry and hot conditions (Figure A.3) due the rapid decrease of available samples in the corners of the three-dimensional probability distribution. Nevertheless, the best estimates (bars in Figures 8 and A.3) show that compound dry and hot extremes contribute to increase yield loss."
* * *
[Figure]

[Figure]

**Fig. 1.** Updated Figure 2

---

## Author Comment (AC4) · 24 Jun 2020

Editor: (1) The three reviewers agree on the clear purpose and structure of the manuscript and are all supportive to publication of this manuscript subject to minor edits. Some refer to justification of (implicit) choices made, some ask for some extra clarification. I would recommend to reply to all review comments that are raised.

Author's Reply (1): Thank you. We have answered to all the review comments individually and in detail.

Editor: (2) For me the most significant comments are:

[Figure]

Additional justification regarding:

- The choice of the 3-month averaging window for the meteorological quantities

- The processing of the negative dependence (why not use a rotated copula, as suggested by a reviewer)

- Availability of the code (and data if that is open)

Author's Reply (2): Thank you. We have addressed these points in detail and we agree to publish the code in a repository.

Editor: (3) Some further explanation concerning:

- The copula parameter theta

- The implication of C being lower than the AIC

- The choice of the marginals in section 2.3

- The reconstruction of the correlation between yield and temperature for wheat 2

- The physical interpretation, not only of the notion that drought and heat alone can give stronger effects than their combination, but in general sense: why would the combination of environmental drivers lead to stronger yield reductions? Is this physiologically explainable?

Author's Reply (3): Thank you. These points where properly addressed. Moreover, in the general sense, the biophysiological explanation for the combination of environmental drivers leading to stronger yield reductions relates with the crop's requirements of water and thermal conditions during the key phenological stage in analysis. The selection of the climate variables during spring corresponds to the reproductive phase of the plant's and when vegetation is photosynthetically more active, and the combined effect of water and heat stress during this period is critical for crop's health leading to yield decrease. During this stage of formation of the grains the compound dry and hot

extremes may accelerate the maturation reducing the size, number and weight of the grains and consequently reducing crop's harvests in quantity and quality (Balla et al., 2011; COPA-COGECA, 2003; Nicolas et al., 1984; Qaseem et al., 2019; Talukder et al., 2014).

In the revised version we will add the following text to the Discussion section:

"Nevertheless, the best estimates (bars in Figures 8 and A.3) show indeed that compound dry and hot extremes contribute to increased yield loss. In the general sense, the biophysiological explanation for the combination of environmental drivers leading to stronger yield reductions relates with the crop's requirements of water and thermal conditions during the key phenological stage in analysis. The selection of the climate variables during spring corresponds to the reproductive phase of the plant's and when vegetation is photosynthetically more active, and the combined effect of water and heat stress during this period is critical for crop's health leading to yield decrease. During this stage of formation of the grains the dry and hot extremes may accelerate the maturation affecting the size, number and weight of the grains and consequently affecting crop's harvests in quantity and quality (Balla et al., 2011; COPA-COGECA, 2003; Nicolas et al., 1984; Qaseem et al., 2019; Talukder et al., 2014).

References:

Balla, K., Rakszegi, M., Li, Z., Békés, F., Bencze, S. and Veisz, O.: Quality of winter wheat in relation to heat and drought shock after anthesis, Czech J. Food Sci., 29(2), 117–128, doi:10.17221/227/2010-cjfs, 2011.

Nicolas, M. E., Gleadow, R. M. and Dalling, M. J.: Effects of drought and high temperature on grain growth in wheat., Aust. J. Plant Physiol., 11(6), 553–566, doi:10.1071/PP9840553, 1984.

Qaseem, M. F., Qureshi, R. and Shaheen, H.: Effects of Pre-Anthesis Drought, Heat and Their Combination on the Growth, Yield and Physiology of diverse Wheat (Triticum

aestivum L.) Genotypes Varying in Sensitivity to Heat and drought stress, Sci. Rep., 9(1), 1–12, doi:10.1038/s41598-019-43477-z, 2019.

Talukder, A. S. M. H. M., McDonald, G. K. and Gill, G. S.: Effect of short-term heat stress prior to flowering and early grain set on the grain yield of wheat, F. Crop. Res., 160, 54–63, doi:10.1016/j.fcr.2014.01.013, 2014.

Editor. (4) Some editing would be recommended regarding:

- The renaming of "cluster" to "region" (as "cluster" does not have a strong geographical association)

- The labeling of heat percentiles in Figure 7.

Author's Reply (4): Thank you. We will change "cluster" to "region" in the revised version and relabel the heat percentiles in Figure 7 properly, as also suggested by the reviews.

---

## Author Response (AR1)

Dear Editorial Board,

Please consider our revised manuscript "Risk of crop failure due to compound dry and hot extremes estimated with nested copulas" by Andreia Ribeiro et al., which we would like to submit for publication in the Biogeosciences journal as an original research article, in the Special Issue "Understanding compound weather and climate events and related impacts".

We truly appreciate the Reviewer's and Editor's feedback about the manuscript, which have been very helpful in improving the manuscript. Below we provide a point-by-point response to the reviews (from the three reviewers and editor) followed by the marked-up manuscript version. Accordingly, the main changes in the manuscript consisted in:

- Replace the word "cluster" to "region"
- Improve labels in Figures 2 and 7
- Clarification regarding the seasons, the vegetative cycle of the crops and the 3-month averaging window of the weather data
- Introducing the copula parameter
- Explain better the estimation of the margins
- Explaining the requirement of positive dependence
- Interpretation of the slightly lower AIC in the case of barley in Region 2
- Explain the reproducibility of the correlation between Tmax and yield
- Improve discussion about uncertainties
- The physiological explanation for the stronger yield reductions under compound dry and hot extremes

Additionally, the authors took the liberty of performing the following changes:

- Clarification about Figure 1
- Shortening of captions in Fig. 6 and Table A1
- Delete two redundant sentences in the abstract

The changes allowed to improve the quality of the paper and in case of publication we will publish the code in http://impecaf.rd.ciencias.ulisboa.pt/.

With my best regards,

Andreia Ribeiro (PhD student)

**Anonymous Referee #1**

This study evaluated the risk of crop failure due to compound dry and hot extremes. A copula model is fitted to estimate the response of crop yield with respect to different dry and hot conditions. This manuscript is well crafted with clear structure.

Reply (0): Thank you for this positive assessment.

A few issues need to be addressed before the potential publication of this study.

(1) Selection of the periods Line 92-93: "We used 3-monthly means of Tmax and 3-monthly means of P during spring". Here the selection is based on the correlation analysis, but not the whole growing season, right? Please justify this period. It is easy to understand this from a statistical perspective. Is this selection still valid from a physical perspective?

Reply (1): Thank you for the question. Given the importance of assessing crop's water and temperature requirements at different moments of the vegetative cycle we conducted a correlation analysis between the yield and the 3-monthly means of precipitation and 3-monthly means of maximum temperature during the whole growing season (approximately from September of year n-1 to June of the year n), as shown in Fig. 2.

The identification of the moment of the vegetative cycle of the highest crop's water and temperature requirements was assessed based on the strongest statistically significant correlation value.

Fig. 2 suggests that the greatest influence of P and Tmax in crop yields is observed during spring (in both regions and cereals) corresponding to the moments in which the vegetation is photosynthetically more active. The effects of water content and high temperatures during middle growth stages of the crop life cycle are in accordance with previous studies (Ferrise et al., 2011; García del Moral et al., 2003; Iglesias and Quiroga, 2007; Ribeiro et al., 2019). Hence, this selection is valid both from the statistical and biophysical point of view.

We have addressed this aspect in the revised manuscript in the Weather data section:

"The vegetative cycle of the winter crops in Spain is mainly driven by precipitation and temperature: sowing occurs around autumn, followed by the vegetative phase in winter, reproductive phase (more photosynthetically active phase) in spring (when vegetation is photosynthetically more active) and crop harvest occurs in the early summer. Therefore, monthly precipitation (P) and monthly maximum temperature (Tmax) were extracted from the Climate Research Unit (CRU) TS4.01 dataset (Harris et al., 2014) spanning the same time period. Given the importance of assessing crop's water and temperature requirements at different moments of the vegetative cycle we conducted a correlation analysis between the annual yields and the 3-monthly means of P and 3-monthly means of Tmax during the whole growing season, as shown in Fig. 2. The identification of the moment of the vegetative cycle of the highest crop's water and temperature requirements was assessed based on the strongest statistically significant correlation value (denoted by filled circles in Fig. 2). Figure 2 suggests that the greatest influence of P and Tmax in

crop yields is observed during spring (MAM in both regions and cereals) corresponding to the reproductive phase of plant development, when vegetation is photosynthetically more active. Therefore, for the remaining analysis we focus on 3-monthly means of Tmax and 3-monthly means of P during spring ($P_{MAM}$ and $Tmax_{MAM}$, respectively), which has also been identified in previous studies as a growth stage sensitive to the effects of water content and high temperatures (Ferrise et al., 2011; García del Moral et al., 2003; Iglesias and Quiroga, 2007; Ribeiro et al., 2019). This selection of climate variables allows to maximize the dependence between climate conditions and yields as also shown by previous work based on the same data (Ribeiro et al., 2019c).”

Reply (3): This means that in the case of barley in Cluster 2, the trivariate copula fits the data slightly better than the two-parameter NAC C(u3, C12(u1,u2)) in terms of AIC, even though the Cramer-von Mises distance is better for the NAC. This could mean that in this case a NAC structure favouring the dependence between yield and precipitation may be less relevant compared to the other clusters and yields. Drought individually seems to play a less dominant role in the compound event, in comparison to the other cereals and regions.

This interpretation would be consistent with our discussion about Figure 8, where barley in Cluster 2 is also the case with the highest difference between drought and compound dry and hot conditions, hence illustrating that here drought is the least dominant driver of crop loss in comparison to the other cereals and regions.

Following the reviewer's suggestion we have added a sentence in the Results section:

"The only exception is barley in Cluster 2 whose AIC of Cθ(u1,u2,u3) is slightly lower than the AIC of Cθ1 (u3,Cθ12 (u1,u2)) (Table 2). This feature may suggest that a structure favouring the dependence between yield and precipitation (u1,u2) may not be as relevant as in the other clusters and yields due to a less dominant role of drought individually in this case. Nevertheless, in terms of Cramer-von Mises distance (Sn) the nested copula is the closer to the empirical trivariate copula. For this reason, we modelled the trivariate joint distribution based on nested Frank copulas for all cases. (…)"

(4) *Code availability. It is more and more common to publish the code used in scientific publications and I strongly recommend the authors considering to publish their code once the paper is accepted.

Reply (4): In case of publication we agree to publish the code in http://impecaf.rd.ciencias.ulisboa.pt/.

**Anonymous Referee #3**

This is a well-written manuscript that investigates the compound effects of precipitation and temperature on crop failure in two provinces in Spain using nested copulas. It contributes to a better understanding of these type of compound events and it therefore deserves publication.

Reply (0): Thank you for this positive assessment.

I have just some minor comments that I recommend the authors to address before publication:

(1) p.4 line 90. Is it monthly daily mean precipitation or monthly accumulated precipitation?

Please specify.

Reply (1): Thanks for recommending the clarification, CRU TS4.01 provides monthly cumulative precipitation, hence we have clarified in the revised manuscript:

"Therefore, monthly ==accumulated== precipitation (P) and monthly maximum temperature (Tmax) were extracted from the Climate Research Unit (CRU) TS4.01 dataset (…)"

(2) p.5 Figure 2. I would highlight the month MAM as it is the choice to of this study to perform the compound event analysis.

Reply (2): We appreciate the suggestion and propose to denote MAM in bold text.

[Figure]

(3) p 7. line 152. I think this Section should explain what the choice for the marginals is

and why.

Reply (3): Thank you for the comment. We used empirical ranks as explained in the methods sections. We have added the following text in the revised manuscript:

The main steps of the trivariate approach used in this study can be summarized as follows (Okhrin and Ristig, 2014). First, the marginal distributions u1, u2 and u3 are estimated non-parametrically by simple ranking, using the empirical distribution functions of the data through the *pobs* function in the R package *copula* (Ivan Kojadinovic and Jun Yan, 2010), a common approach for copula modelling."

**Bart van den Hurk (Editor)**

The three reviewers agree on the clear purpose and structure of the manuscript and are all supportive to publication of this manuscript subject to minor edits. Some refer to justification of (implicit) choices made, some ask for some extra clarification. I would recommend to reply to all review comments that are raised.

Reply (1): Thank you. We have answered to all the review comments individually and in detail.

For me the most significant comments are:

Additional justification regarding:

• The choice of the 3-month averaging window for the meteorological quantities

• The processing of the negative dependence (why not use a rotated copula, as suggested by a reviewer)

• Availability of the code (and data if that is open)

Reply (2): Thank you. We have addressed these points in detail and in case of publication we agree to publish the code in http://impecaf.rd.ciencias.ulisboa.pt/.

Some further explanation concerning:

• The copula parameter {theta • The implication of C being lower than the AIC

• The choice of the marginals in section 2.3

• The reconstruction of the correlation between yield and temperature for wheat 2

• The physical interpretation, not only of the notion that drought and heat alone can give stronger effects than their combination, but in general sense: why would the combination of environmental drivers lead to stronger yield reductions? Is this physiologically explainable?

Reply (3): Thank you. These points where properly addressed. Moreover, in the general sense, the biophysiological explanation for the combination of environmental drivers leading to stronger yield reductions relates with the crop's requirements of water and thermal conditions during the key phenological stage in analysis. The selection of the climate variables during spring corresponds to the reproductive phase of the plant's and when vegetation is photosynthetically more active, and the combined effect of water and heat stress during this period is critical for crop's health leading to yield decrease. During this stage of formation of the grains the compound dry and hot extremes may accelerate the maturation reducing the size, number and weight of the grains and consequently

reducing crop's harvests in quantity and quality (Balla et al., 2011; COPA-COGECA, 2003; Nicolas et al., 1984; Qaseem et al., 2019; Talukder et al., 2014).

In the revised version we have added the following text to the Discussion section:

"Nevertheless, the best estimates (bars in Figures 8 and A.3) show indeed that compound dry and hot extremes contribute to increased yield loss. In the general sense, the biophysiological explanation for the combination of environmental drivers leading to stronger yield reductions relates with the crop's requirements of water and thermal conditions during the key phenological stage in the analysis. The selection of the climate variables during spring corresponds to the reproductive phase of the plant's and when vegetation is photosynthetically more active, and the combined effect of water and heat stress during this period is critical for the crop's health leading to yield decrease. During this stage of formation of the grains the dry and hot extremes may accelerate the maturation affecting the size, number and weight of the grains and consequently affecting the crop's harvests in quantity and quality (Balla et al., 2011; COPA-COGECA, 2003; Nicolas et al., 1984; Qaseem et al., 2019; Talukder et al., 2014).

Balla, K., Rakszegi, M., Li, Z., Békés, F., Bencze, S. and Veisz, O.: Quality of winter wheat in relation to heat and drought shock after anthesis, Czech J. Food Sci., 29(2), 117–128, doi:10.17221/227/2010-cjfs, 2011.

Nicolas, M. E., Gleadow, R. M. and Dalling, M. J.: Effects of drought and high temperature on grain growth in wheat., Aust. J. Plant Physiol., 11(6), 553–566, doi:10.1071/PP9840553, 1984.

Qaseem, M. F., Qureshi, R. and Shaheen, H.: Effects of Pre-Anthesis Drought, Heat and Their Combination on the Growth, Yield and Physiology of diverse Wheat (Triticum aestivum L.) Genotypes Varying in Sensitivity to Heat and drought stress, Sci. Rep., 9(1), 1–12, doi:10.1038/s41598-019-43477-z, 2019.

Talukder, A. S. M. H. M., McDonald, G. K. and Gill, G. S.: Effect of short-term heat stress prior to flowering and early grain set on the grain yield of wheat, F. Crop. Res., 160, 54–63, doi:10.1016/j.fcr.2014.01.013, 2014.

Some editing would be recommended regarding:

• The renaming of "cluster" to "region" (as "cluster" does not have a strong geographical association)

• The labeling of heat percentiles in Figure 7.

Reply (4): Thank you. We have modified "cluster" to "region" in the revised version and relabelled the heat percentiles in Figure 7 properly, as also suggested by the reviews.

[revised manuscript text omitted]